# Assessment of hepatic function, perfusion and parenchyma attenuation with indocyanine green, ultrasound and computed tomography in a healthy rat model: Preliminary determination of baseline parameters in a healthy liver

Victor Lopez-Lopez[1]*, Nuria Garcia-Carrillo[2], Diego de Gea[3], Lidia Oltra[4], Carlos Alberto González-Bermúdez[3], Guillermo Carbonell[5], Roberto Brusadin[1], Asunción Lopez-Conesa[1], Ricardo Robles-Campos[1]

1 Department of Surgery, HBP Unit, Virgen de la Arrixaca University Clinical Hospital, University of Murcia, (IMIB-Arrixaca), Murcia, Spain, 2 Laboratory Animal Service, Core Facilities University of Murcia, Murcia, Spain, 3 Faculty of Medicine, University of Murcia, (IMIB-Arrixaca), Murcia, Spain, 4 Department of Physiology, Faculty of Medicine, University of Murcia, (IMIB-Arrixaca), Murcia, Spain, 5 Department of Radiology, Virgen de la Arrixaca University Clinical Hospital, University of Murcia, (IMIB-Arrixaca), Murcia, Spain

* victorrelopez@gmail.com

## Abstract

### Background

Defining reference intervals in experimental animal models plays a crucial role in pre-clinical studies. The hepatic parameters in healthy animals provide useful information about type and extension of hepatic damage. However, in the majority of the cases, to obtain them require an invasive techniques. Our study combines these determinations with dynamic functional test and imaging techniques to implement a non-invasive protocol for liver evaluation. The aim of the study was to determine reference intervals for hepatic function, perfusion and parenchyma attenuation with analytical and biochemical blood parameters, indocyanine green, ultrasound and computed tomography in six healthy SD rats.

### Methods

Six males healthy SD rats were followed for 4 weeks. To determine hepatic function, perfusion and parenchyma attenuation analytical and biochemical blood parameters, indocyanine green, ultrasound and computed tomography were studied. Results were expressed as Means ± standard error of mean (SEM). The significance of differences was calculated by using student t-test, p < 0.05 was considered statistically significant.

**Data Availability Statement:** All relevant data are within the manuscript and its Supporting information files.

**Funding:** The author(s) received no specific funding for this work.

**Competing interests:** The authors have declared that no competing interests exist.

## Results

Indocyanine green clearance 5 and 10 minutes after its injection was 80.12% and 96.59%, respectively. Approximate rate of decay during the first 5 minutes after injection was 38% per minute. Hepatic perfusion evaluation with the high-frequency ultrasound was related to cardiovascular hemodynamic and renal perfusion. Portal area, hepatic artery resistance index, hepatic artery and portal peak systolic velocity and average between hepatic artery and porta was $3.41 \pm 0.62$ mm$^2$, $0.57 \pm 0.04$ mm$^2$/s, $693.24 \pm 102.53$ mm$^2$/s, $150.72 \pm 17.80$ mm$^2$/s and $4.82 \pm 0.96$ mm$^2$/s, respectively. Heart rate, cardiac output, left renal artery diammetre and renal blood flow were $331.01 \pm 22.22$ bpm, $75.58 \pm 8.72$ mL/min, $0.88 \pm 0.04$ mm$^2$ and $13.65 \pm 1.95$ mm$^2$/s. CT-scan hepatic average volume for each rat were $21.08 \pm 3.32$, $17.57 \pm 2.76$, $14.87 \pm 2.83$ and $13.67 \pm 2.45$ cm$^3$ with an average attenuation coefficient of $113.51 \pm 18.08$, $129,19 \pm 7.18$, $141,47 \pm 1.95$ y $151,67 \pm 1.2$ HU.

## Conclusion

Indocyanine green and high-frequency ultrasound could be used in rats as a suitable marker of liver function. Computed tomography, through the study of raw data, help to characterize liver parenchyma, and could be a potential tool for early detection of liver parenchymal alterations and linear follow-up of patients. Further studies in rats with liver disease are necessary to verify the usefulness of these parameters.

## Introduction

Study of hepatic parenchyma structure, functionality and perfusion assessment are part of liver surgery planning. Currently include liver biopsy, together with plasma biochemical parameters and Indocyanine Green Clearance (IGC) determination, or image analysis: ultrasonography, eco-doppler and computer tomography (CT) among other imaging techniques. These parameters can provide useful information about type and extension of hepatic damage [1, 2]. However, it is necessary to combine these determinations with dynamic functional test and imaging techniques to implement a non-invasive protocol for liver evaluation.

Dynamic functional tests are able to monitorize current hepatocellular function. One of the most commonly used is plasma clearance of Indocyanine Green (ICG) [3]. Serial determination of plasma ICG concentration by spectrophotometry provides a non-invasive, fast and easy method to determine hepatocellular function and hepatic perfusion in hepatic chronic diseases, critical ill patients, liver transplantation procedures or hepatic lobe resection planning [3, 4]. With regard to imaging techniques, ultrasonography is currently considered a first-line method for the study of focal and diffuse liver diseases. Ultrasounds also provides an accessible and non-invasive technique for monitoring dynamic hepatic perfusion. In this regard, intrahepatic vascular modification as well as portal hypertension have been associated to chronic liver diseases [5–7]. These alterations will have consequences in cardiac and renal hemodynamic [8, 9]. Together with ultrasonography, the inclusion of high-resolution CT-scanners and helical detectors in liver diseases studies, allows to obtain multiplanar imaging and tridimensional hepatic reconstruction by computing [10]. In this process, the obtained image is divided in cubic voxels, allowing to analyse liver attenuation value in concrete portions of liver parenchyma, which could correspond to defined lesions.

Translational medicine can be defined as the transfer of non-clinical findings into clinical applications for a better understanding of human diseases. In this respect, animal models play a crucial role in pre-clinical studies. Due to their ease of handling, accessibility, size, possibility of obtaining different number of samples and reproducibility of studies, Sprague Dawley (SD) rats are considered as an ideal experimental model for liver diseases and surgery [11]. However, these studies entail multiple animal sacrifices to reach an optimal experimental sample size, increasing costs and rising controversy about animal welfare in research. With the aim of minimizing these limitations, the implementation of 3Rs principles (refinement, replacement and reduction) needs to be considered [12]. These principles include the reduction of animal use, which is limited by the number of animals needed to be statistically acceptable. According to the mentioned above, the present study is aimed to determine hepatic biochemical profile, clearance of ICG, hepatic perfusion indexes measured by ultrasound, and average liver attenuation HU CT values in healthy SD rat's different liver segments. They will allow to establish a preliminary range of references values for its future comparison with the ones registered in liver diseases or disorders, helping to minimize the number of animals needed in liver animal research without risking statistical significance.

## Materials and methods

### Animal and ethics

Six males SD rats aged 12 weeks were purchased from the University Animal Center REGA-ES300305440012 (Murcia, Spain) and maintained in individual cages under controlled environmental conditions: room temperature $23 \pm 3°C$, $55 \pm 10\%$ relative humidity and 12 h light/dark cycles. Food and water were provided ad-libitum. Inclusion criteria was no previous history or signs of hepatopathy, as well as analytical and biochemical blood parameters in normal range [13]. In order to discard liver structural or vascular abnormalities, an abdominal High-Frequency Ultrasound (HFU) exploration was performed at the beginning of the study, following the protocol described below. Animal were followed for 4 weeks, controlling weight in order to detect stress. Weight measurements during the experiments are shown in Table 2. Animals were euthanized at 16 weeks of age.

Experiments were designed and conducted according to the "Guide for the Care and Use of Laboratory Animals" (8th edn) [14] and European Directive 2010/63/EU [15]. Project was approved by the University of Murcia's "Institutional Animal Care and Use Committee" (CEEA) in process number A13201001.

### Blood biochemical analysis

Animals were initially sedated using isoflurane vapor (3% induction; 2–2.5% maintenance). Blood samples were obtained from lateral tail-vein. For blood vessels dilation, tail was immersed in hot water (40°C) for 3 min. After vasodilation, a 28-Gauge needle attached to a catheter was inserted 2–3 cm away from the tip of the tail at an angle of approximately 20°. Catheter was flushed with saline solution plus heparin (1–2 UI/mL) between each blood sample extraction [16, 17].

Blood samples were handled to obtain serum. Serum Biochemical parameters (total proteins, albumin, globulin, total cholesterol, Alkaline Fosfatase (AP), Gamma Glutamyl Transpherase (GGT), Serum Alanine Aminotransferase (ALT), Aspartate Aminotransferase (AST), total bilirubin, bile acids, urea, creatinine, Paroxonase 1 (PON 1) and prothrombin time) were obtained using a Roche Cobas 8000 (Roche Diagnostics, Mannheim, Germany).

## ICG plasma clearance determination

A 2.5 mg/mL ICG dilution in sterile water was injected via catheter to each animal. After injection, seriated blood samples were collected at 1, 5 and 10 min. Each sample was conditioned diluting 150 μL of serum in 750 μL of a solution of 1% Bovine serum albumin and 0.9% NaCl. ICG clearance semi-logarithmic curve was determined by spectrometer (PowerWave XS microplate spectrophotometer®, BioTek, Winooski, VT, USA) at 850 nm. ICG clearance constant (elimination rate constant) (ICG-K) was calculated as the curve slope (first derivative). Plasma ICG rate of decay per minute (R) during the exponential decay phase was calculated as: $R = I–d$ and $Log\ d = (Log\ C2—log\ C1)/t2—t1$. C2 and C1, were plasma ICG concentrations at time 1 (injection of ICG) and time 2 (5 minutes). Plasma ICG clearance was expressed as %/min.

## High-Frequency Ultrasound (HFU) examination

After being anesthetized, the abdomen was shaved and further cleaned with a chemical hair remover to minimize ultrasound attenuation was previously described by Chen et al. [18]. Study was performed using a commercial HFU system (Visual Sonics Vevo 3100®, Toronto, Ontario, Canada) connected to a MX250 (axial resolution: 50 μm; frequency: 25 MHz) and MX400 (axial resolution: 75 μm; frequency: 40 MHz). Ultrasound gel was laid on the skin as coupling fluid. Rats were explored at 12 and at 16 weeks of age, to guarantee inclusions criteria [13], by two experienced researchers from the University Animal Center REGA-ES300305440012 (Murcia, Spain). Organs and vessels were explored with B and Doppler mode. Pulse repetition rate frequency was set from 4 to 48 KHz, with a Doppler sample size between 0,25–0,5 mm and an insonation angle lower than 60 º. Images analysis was performed with Vevo Lab 3.0.0 Software (Fujifilm-Visualsonics®, Toronto, Ontario, Canada).

Liver was examined with B-mode to assess surface, contour, parenchyma echostructure and echogenicity, as well as to define main blood vessels. Bidimensional planes were obtained by optimizing gain compensation at 35dB. Portal vein was transversally sectioned at the portal space, obtaining the average area size from 3 different consecutive measurements. In order to stablish liver hemodynamic and perfusion, a Doppler-mode study was conducted. Transductor was placed longitudinally on portal space (portal vein and hepatic artery) and on central axis of liver (cava vein), obtaining main vessels flux velocity. Higher peak on systolic wave was considered as Peak Systolic Velocity (PSV), whereas lower point between systolic peaks was defined as "Ending Diastolic Velocity" (EDV). For each animal, Integral Velocity in Time (VTI) (area under the curve), as well as average PSV and EDV where calculated from 5 consecutive cardiac cycles. Vascular resistance index was calculated as portal congestion index (PCI = portal average area/average PSV) and arterial/portal ratio (A/P = average hepatic artery/average portal).

Hepatic perfusion analysis was completed with the study of cardiovascular hemodynamic and renal perfusion. As it has been previously explained, both organs are directly related to hepatic diseases [8, 9]. With this purpose, heart was examined in B-mode. Data were transferred to an ultrasound image workstation for analysis (Vevo LAB 3.1.1®). The highest point of the systolic waveform was defined as PSV, whereas the lower point of the diastolic waveform was defined as EDV. Both PSV and EDV were measured from at least five consecutive cardiac cycles. VTI was obtained by outlining five consecutive heartbeats cycles, calculating the integral under the resulting curve. Time-average velocity (TAV) was measured considering the heartbeat cycles by ultrasound system.

For cardiac hemodynamic, B and M-mode echocardiographic evaluations were performed using a 25 MHz transducer. B-mode was activated to visualize the heart structure.

Measurement of stroke volume (SV), HR and CO were obtained from at least three consecutive cardiac cycles in M-mode. For renal hemodynamic, blood flow was measured in left kidney using a 40 MHz transducer. B-mode was activated to visualize the renal artery. Its diameter was measured by tracing a line between the internal opposite sides of the artery wall in two frozen images. Average arterial diameter was obtained from five consecutive measurements. RBF was calculated from the following formula: RBF = HR x VTI x $\pi r^2$, where r is the vessel radius.

## Micro-CT technique

After euthanization of animals (16 weeks of age), livers were extracted and maintained in plastic carriers with physiological serum at room temperature. Micro-CT scan was performed ex-vivo in order to get a better definition and to allow a better multiplanar reconstruction. Total volume was expressed on $cm^3$ and hepatic density was expressed in HU. Samples were imaged, using the preclinical trimodal scanner De Albira SPECT/PET/CT (Bruker [R] Corporation, Karlsruhe, Germany) at the Preclinical Imaging Facilities of the University of Murcia, following the methodology previously described [19, 20]. The X-ray source was set to 200 microamps (mA) and a voltage of 45 peak kilovoltage (kVp), using a 0.5 mm aluminum filter to harden the beam. A digital flat-screen X-ray detector (Bruker [R], Karksruhe, Germany) with 2,400 x 2,400 pixels and a 70 x 70 $mm^2$ FOV were used to capture 600 0.2 $mm^3$ voxel projections. Images were reconstructed in the three orthogonal planes (transversal, coronal and sagittal) by applying the filtered back projection (FBP) algorithm via Albira Suite 5.0 Reconstructor (Bruker [R], Karksruhe, Germany). Hepatic attenuation (HU) for each hepatic segment and average value were determined segmenting Volume of Interest (VOIs) of 8$mm^3$ size over the CT image. Each VOI contained more than 4,900 voxels obtaining the mean density value in HU. All measurements were performed by ensuring the exclusion of macroscopic vessels, using AMIDE post-processing software[R] (University of California, Los Angeles). Density values obtained from liver were used as reference to perform the *in-situ* volume analysis of each whole liver. When density values were properly adjusted, automatic segmentation was carried out with AMIDE software[R], using automated t pathway [21]. 3D isocontour of VOIs was selected to outline the liver. Volumetric segmentation of the liver was performed using a semi-automated tool by HU thresholding. The average tissue density value for each voxel was quantified and transformed into different grey levels by the Hounsfield Units (HU) scale, ranging from -1000 HU (air) to +1000 HU (dense bone) [21]. 0 HU was fixed as lower limit in order to exclude fat tissue and to include non-altered and functional hepatic tissue. Higher limit was pre-set at 120 HU. With these settings, a total of 8,411,177 voxels were approximately obtained from each 3D-Isocontour. Volumetric values ($mm^3$) were acquired with the AMIDE software to determine the proportion (%) of each segmented structure related to the total volume.

## Statistical analysis

Descriptive statistical analysis was performed using SPSS version 24.0. Median and range, as well as mean and standard deviation were calculated for quantitative variables. Frequency distribution was used for qualitative variables.

## Results

With the purpose of determining normality range values for functional and structural hepatic analysis techniques, static and dynamic laboratory tests (blood biochemical analysis and ICG plasma clearance), as well as image analysis (HFU and micro-CT) were performed.

**Table 1. Rats weight evolution during the study (from 12 to 16 weeks of age).** Each animal was weighed 4 times during the study.

| Week | Rat 1 Weight (g) | Rat 1 W.G. per measure (%) | Rat 2 Weight (g) | Rat 2 W.G. per measure (%) | Rat 3 Weight (g) | Rat 3 W.G. per measure (%) | Rat 4 Weight (g) | Rat 4 W.G. per measure (%) | Rat 5 Weight (g) | Rat 5 W.G. per measure (%) | Rat 6 Weight (g) | Rat 6 W.G. per measure (%) | Average weight (μ ±SD) |
|---|---|---|---|---|---|---|---|---|---|---|---|---|---|
| 1 | 390.70 | - | 369.00 | - | 367.40 | - | 328.10 | - | 355.20 | - | 413.70 | - | 370.68 ±29.39 |
| 2 | 390.30 | -0.10 | 367.30 | -0.46 | 372.20 | 1.31 | 340.00 | 3.63 | 350.50 | -1.32 | 427.20 | 3.26 | 374.58 ±31.15 |
| 3 | 406.70 | 4.20 | 376.00 | 2.37 | 392.00 | 5.32 | 353.90 | 4.68 | 370.20 | 5.62 | 435.90 | 2.04 | 389.45 ±28.77 |
| 4 | 418.20 | 2.83 | 385.30 | 2.47 | 408.50 | 4.21 | 372.00 | 4.52 | 383.20 | 3.51 | 454.40 | 4.24 | 403.60 ±30.22 |
| Final W.G. (%) | 7.04 | | 4.42 | | 11.19 | | 13.38 | | 7.88 | | 9.84 | | 8.88 |

**Weight Gain (W.G.) per measure %** has been calculated respect the previous measurement. **Final Weight Gain (W.G.) %** has been calculated as weight difference between week 1 and week 4.

Weight evolution during the experiment has been presented in Table 1. Average initial weight (12 weeks of age) was 370.68±29.39 gr, with an average final weight gain of 8.88% at 16 weeks of age. As can be seen in Table 1, at the end of the second week (measure 2), three specimens experienced a slight weight loss (-0.10%, -0,46% and -1.32). In these animals, final weight gain (%) (end of the experiment) was respectively lower (7.04%, 4.42% and 7.48%) than for the other specimens (11.19%, 13.38% and 9.84%). No relation between final weight gain (%) and initial weight loss was found.

Average blood biochemical analysis results are shown in Table 2. All the parameters showed a slight oscillation in the reference ranges along the study. In the case of prothrombin time, its values drastically decreased from 74.30 ± 9.51 s at week 2, to 18.83 ± 1.47 and 17.73 ± 1.36 s at week 3 and 4, respectively. This finding was attributed to a problem in the laboratory management of the first determinations, so prothrombin time for 13 and 14 weeks of age were discarded.

ICG plasma clearance is commonly used as a marker of liver function and perfusion in both, liver peri-operative assessment and critical patients. In this regard, ICG plasma clearance results are shown in Table 3. As can be seen, after 5 and 10 minutes from its injection, ICG clearance was 80.12% and 96.59% respectively (Fig 1). ICG in plasma of healthy SD rats exponentially decayed for the first 5 minutes after injection. This phase was followed by a deceleration, reaching minimum values at 10 minutes. Approximate rate of decay during the first 5 minutes after injection was 38% per minute (calculated from average ICG concentration values).

HFU study was performed in order to assess liver structure and perfusion. As hepatic perfusion is directly related to cardiovascular hemodynamic and renal perfusion, both were included in the HFU study. In Fig 2A, images for hepatic perfusion evaluation (longitudinal section of portal vein and hepatic artery) are shown. Portal vein showed a hepatopetal flow with a laminar spectrum distribution, in-phase with respiratory movements. Hepatic arteries (b), also showed a hepatopetal flow, with a monophasic spectrum in which, systolic peaks and slow diastolic fall were identified. Hemodynamic study images are presented in Fig 2B. As can be seen, inferior cava vein showed a hepatopetal flux and a multiphasic spectrum. Cardiac

**Table 2. Blood biochemical analysis and published reference values for biochemical analytes in male Sprague-Dawley rats.**

| Biochemical Parameters | Week 1 (μ ± SD) | Week 2 (μ ± SD) | Week 3 (μ ± SD) | Week 4 (μ ± SD) | Lillie et al., (1996) [24] Average±SD | Petterino & Argentino (2006) [25] Ref. Interval | Han et al., (2010) [22] Ref. interval | He et al., (2017) [23] Ref. interval |
|---|---|---|---|---|---|---|---|---|
| Rat age | 12–16 weeks | | | | 5–7 weeks | 13 weeks | 13 weeks | 9 weeks |
| Total Protein (g/dL) | 4.58 ± 0.39 | 4.93 ± 0.45 | 4.84 ± 0.37 | 4.59 ± 0.24 | 5.85 ± 0.23 | 6.5–8.1 | 5.68–9.25 | 5.11–6.45 |
| Albumin (g/dL) | 1.88 ± 0.52 | 1.02 ± 1 | 2.32 ± 0.21 | 2.11 ± 0.12 | 3.08 ± 0.11 | 2.9–4.1 | 2.83–4.05 | 2.69–3.46 |
| Globulins (g/dL) | 2.70 ± 0.20 | 3.91 ± 0.67 | 2.52 ± 0.20 | 2.48 ± 0.18 | - | 4.83–5.13 | 2.24–2.89 | - |
| T. Cholesterol (mmol/L) | 2.27 ± 0.26 | 1.99 ± 0.21 | 1.95 ± 0.07 | 1.89 ± 0.13 | 1.01 ± 0.22 | 1.9–4.6 | 1.86–5.34 | 0.68–1.77 |
| ALP (IU/L) | 337.05 ± 91.12 | 360.27 ± 33.10 | 271.30 ± 54.73 | 238.63 ± 40.04 | 290 ± 63 | 131.6–459 | 58.4–180.4 | - |
| GGT (IU/L) | UDL* | UDL* | UDL* | UDL* | UDL* | UDL* | - | - |
| AST (IU/L) | 62.68 ± 10.18 | 51.70 ± 6.20 | 54.92 ± 6.42 | 44.50 ± 3.24 | 78.1 ± 13.0 | 56.1–201.8 | 64.1–168.1 | 60–139 |
| ALT (IU/L) | 36.62 ± 6.83 | 34.25 ± 3.89 | 37.77 ± 3.87 | 31.42 ± 3.74 | 28.9 ± 5 | 34.9–218.1 | 30.8–73.4 | - |
| BUN (mmol/L) | 5.12 ± 0.52 | 5.37 ± 0.25 | 5.06 ± 0.64 | 4.94 ± 0.64 | 9.46 ± 0.84 | 10.8–34.4 | 12.1–26.1 | 4.32–8.97 |
| Bilirubin (mg/dL) | 1.74 ± 0.71 | 0.99 ± 0.35 | 1.18 ± 0.46 | 1.31 ± 0.36 | 1.4 ± 0.6 | 0.0–0.3 | 0.07–0.29 | - |
| Bile acids (µmol/L) | 23.88 ± 18.53 | 41.40 ± 22.35 | 24.51 ± 7.48 | 25.98 ± 10.81 | - | - | - | - |
| PON1 (IU/L) | 2 ± 1.51 | 2.85 ± 0.21 | 3.12 ± 0.26 | 2.90 ± 0.32 | - | | - | - |
| Prothrombin Time (s) | 41.57 ± 29.44 | 74.30 ± 9.51[d] | 18.83 ± 1.47 | 17.73 ± 1.36 | - | 10.3–18.2 | 8.1–18.3 | - |
| Creatinin (µmol/L) | - | - | - | - | 47.6 ± 7.4 | 35.4–79.6 | 22.1–73.37 | 32.36–47.90 |

*UDL: Under Detection Limit

[d]Values discarded due to an error in the laboratory quantification.

motility and volumes were evaluated in M and B-mode. Determined values for perfusion and hemodynamic assessment are compiled in Table 4.

Micro-CT images of rat's liver are presented in Fig 3. Hepatic average volume for all rats were 21.08±3.32, 17.57±2.76, 14.87±2.83 and 13.67±2.45 cm$^3$ with an average attenuation coefficient of 113.51±18.08, 129,19±7.18, 141,47±1.95 y 151,67±1.2 HU (Fig 4). Estimated

**Table 3. Indocyanine green plasma clearance (ICG) rate.**

| | ICG plasma concentration (µg/mL) | | |
|---|---|---|---|
| | min 1 | min 5 | min 10 |
| Rat 1 | 128.1±55.618 | 19.45±2.28 | 5.41±3.32 |
| Rat 2 | 161.47±65.82 | 28.58±9.23 | 5.75±3.60 |
| Rat 3 | 160.63±27.68 | 33.14±6.95 | 7.44±0.52 |
| Rat 4 | 189.16±51.05 | 38.13±2.86 | 3.47±0.05 |
| Rat 5 | 165.91±23.91 | 47.51±8.82 | 2.90±0.76 |
| Rat 6 | 130.65±10.21 | 19.24±3.54 | 6.91±0.99 |
| Average ICG Concentration (µg/mL) | 155.99±23.11 | 31.01±11.01 | 5.31±1.82 |
| Average Clearance % | 0.00 | 80.12 | 96.59 |

µg/mL: microgram/mililitre; min: minute.

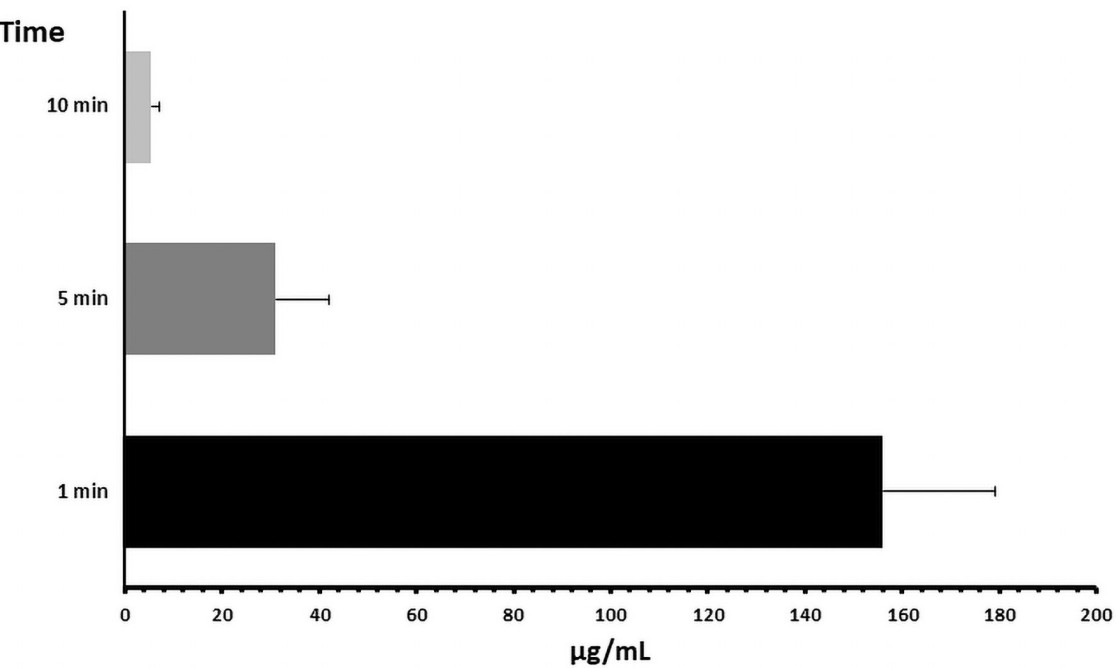

**Fig 1. ICG plasma clearance at 1, 5 and 10 min.**

hepatocyte attenuation coefficient were in a range of 134–137 HU. Values of 150–200 HU were assigned to hepatic capsule cells.

## Discussion

Considering that SD rats are one of the most widely animal models used in liver translational medicine, the purpose this study is to establish healthy SD rats' reference values. In order to obtain them, we have selected those complementary test related to liver function and which, once established as representative of the species, would serve in the future for the prediction of responses in different pathological situations. This study was conducted with the purpose of establishing preliminary reference values of healthy rats.

Due to the lack of validated reference values and the impossibility of stablish confidence intervals when designing a new project, researches are forced to include a higher number of animals in control group than in test group.

With the aim of establishing reference values for hematological, biochemical and physiological healthy males SD rats parameters, the present study was designed. This should reduce the number of experimental animals needed in future researches, helping to implement Russell and Burch 3Rs reduction principle.

According to Table 2, during the first week (12 week of age), some animals (rats 1, 2 and 5) experienced a weight loss which varied from -0.10% to -1.32%. For the rest of specimens, a slight weight gain (from 1.31% to 3.63%) was observed. Together with the individual characteristics of each animal, these differences can be attributed to initial environmental and handling adaptation [26, 27]. This could explain why animals' weight increased from second week to the end of the experiment (16 weeks of age), reaching an average final weight of 403.60 ±30.22 g. On the contrary, first week could be considered as adaptation period. Final weight

A)

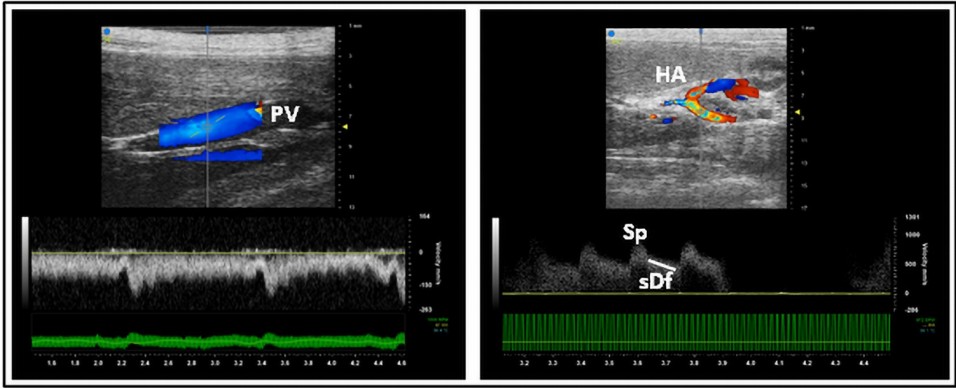

B)

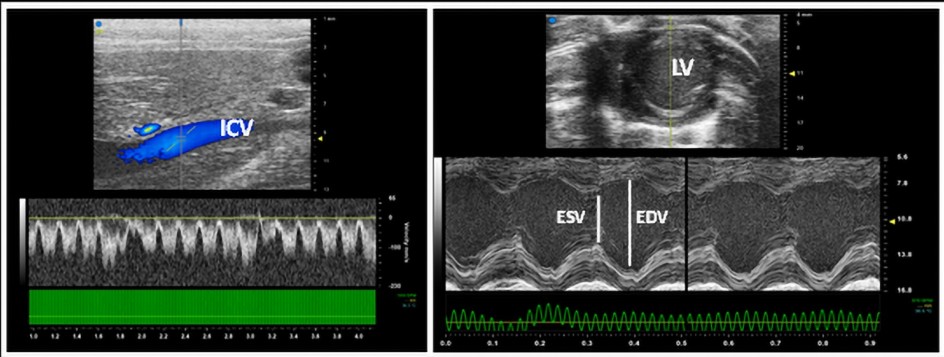

**Fig 2. High-Frequency Ultrasound (HFU) examination for hepatic and cardiovascular hemodynamic assessment.**
A) Abdominal HFU study for hepatic perfusion analysis. Longitudinal Colour and Spectral Doppler sections of Portal
Vein (PV) and Hepatic Artery (HA) are presented. Systolic peak (Sp) and slow Diastolic fall (sDf) have been indicated.
B) HFU study for cardiovascular hemodynamic assessment. Colour and Spectral Doppler transversal sections of
Inferior Cava Vein (ICV). B-Mode and M-mode images of transverse section of Left Ventricle (LV). Motility, End
Systolic and End Diastolic Volumes (ESV and EDV respectively) were evaluated in M-mode. PV: porta vein, HA:
hepatic artery.

gain ranged between 4.42% and 13.38%. No relation between final weight gain and initial
weight loss was found, being these differences attributed to individual characteristics.

With regard to blood biochemical analysis results, different published studies have been
conducted with the aim of establishing reference biochemical and haematological values for
healthy SD rats [28, 29]. Reference values from these studies have been compiled in Table 1. As
can be seen, most of the analysed values are similar to those indicated in the consulted pub-
lished reference values or intervals. On the contrary, obtained values for total cholesterol,
GGT, bilirubin, ALP and prothrombin time, showed some discrepancies with the published
results. No reference values for bile acids or PON1 in SD rats were found in the literature.

Results of plasma ICG clearance as a non-invasive marker of liver perfusion are displayed
in Table 3. As it has been previously explained, ICG plasma concentration drastically decayed
during the first 5 minutes after injection (exponential phase), with an average clearance of
80.12% and a decay rate of 38% per minute. This is something considered when adapting ICG
clearance tests from humans to SD rats. As it was described by Cherrick et al. [30], plasma ICG
clearance in healthy humans show an exponential decay during the first 20 minutes after

**Table 4. Cardiovascular hemodynamic, hepatic and renal perfusion results obtained by HFU.**

| | Rat 1 | Rat 2 | Rat 3 | Rat 4 | Rat 5 | Rat 6 | Average measurement (μ±SD) |
|---|---|---|---|---|---|---|---|
| Portal area (mm$^2$) | 3.09 | 3.60 | 3.53 | 2.67 | 3.12 | 4.48 | 3.41 ± 0.62 |
| Portal PSV (mm$^2$/s) | 131.25 | 161.00 | 148.60 | 176.20 | 130.55 | 156.75 | 150.72 ± 17.80 |
| HA PSV (mm$^2$/s) | 864.37 | 687.47 | 643.69 | 627.48 | 581.00 | 755.47 | 693.24 ± 102.53 |
| HA EDV (mm$^2$/s) | 424.33 | 266.29 | 309.93 | 262.63 | 232.95 | 304.78 | 300.15 ± 67.24 |
| HA IVT (mm/s) | 110.03 | 82.37 | 79.06 | 73.06 | 60.09 | 84.23 | 81.47 ± 16.47 |
| HA RI (mm$^2$/s) | 0.52 | 0.59 | 0.51 | 0.57 | 0.62 | 0.61 | 0.57 ± 0.04 |
| PCI (mm$^2$/s) | 0.02 | 0.02 | 0.02 | 0.02 | 0.03 | 0.03 | 0.023 ± 0.005 |
| A/P (mm$^2$/s) | 6.64 | 4.39 | 4.44 | 3.84 | 4.64 | 4.98 | 4.82 ± 0.96 |
| HR (bpm) | 331.50 | 303.13 | 321.08 | 316.71 | 362.13 | 351.50 | 331.01 ± 22.22 |
| CO (mL/min) | 79.64 | 67.09 | 74.03 | 67.91 | 74.12 | 90.71 | 75.58 ± 8.72 |
| LRAD (mm) | 0.91 | 0.90 | 0.81 | 0.85 | 0.92 | 0.90 | 0.88 ± 0.04 |
| Renal IVT (mm/s) | 64.37 | 71.04 | 67.21 | 68.91 | 63.59 | 72.01 | 67.86 ± 3.44 |
| RBF (mm$^2$/s) | 14.07 | 13.27 | 11.01 | 12.07 | 15.18 | 16.28 | 13.65 ± 1.95 |

mm: millimetre, s:seconds, bpm: beats per minute, mL: millilitre, min: minutes, HA: hepatic artery, PSV: peak systolic velocity, EDV: ending diastolic velocity, IVT: integral velocity in time, RI: resistance index, PCI: portal congestion index, A/P: average hepatic artery /average portal, HR: heart rate, CO: cardiac output, LRAD: left renal artery diameter, RBF: renal blood flow.

injection, with a decay rate of 18.5%. In contrast, and according to our results, plasma ICG clearance in SD rats should be measured in a shorter period of time than in humans, in order to analyse its behaviour during the exponential phase. According to these authors, pharmaco-kinetics of ICG in SDs' plasma corresponds to a "one-compartment model" with three stages and no absorption phase. In this way, after injection (stage 1) ICG rapidly distributed and stabilized in plasma (stage 2). These two phases only take a few seconds. After blood stream stabilization, a 3rd stage found, in which ICG is filtered from plasma by liver. In this 3rd stage, an apparent exponential decrease is described, with a decay rate of 41 ± 5% per minute. Our results agree with Dorshow et al. [31], showing a similar behaviour and a similar decay rate per minute. These considerations should be taken into account when analysing plasmatic ICG clearance.

In parallel, liver structure and perfusion, together with cardiovascular hemodynamic and renal perfusion were characterized by HFU (Table 4). Ultrasound is a widely used tool, being validated in both: human and animal models. In 2010 Lessa et al. [32] demonstrated that the use of ultrasound images in the diagnosis of rodents' liver disease is feasible and efficient, describing a homogeneous liver parenchyma. In this study, variation in the calibre of portal vein and portal blood pressure were related to the development of hypertension. In the same way, D´Souza et al. [33], demonstrated, that this technique allowed to obtaining similar results as liver anathomopatological studies, avoiding the drawbacks of biopsy. Furthermore, HFU is able to explore different hepatic segments being able to monitoring liver cirrhosis development and chronic liver disease evolution. In the present study, liver structure and perfusion, together with cardiovascular hemodynamic and renal perfusion were characterized by ultrasound. In addition, the standardization of portal and arterial flow measurements in SD rat models are of great interest, due to their scarce representation in the literature as well as to their great applicability in liver preclinical studies.

In parallel, liver anatomical and functional study was also performed using micro-CT. In our study, "raw data" have been analyzed. These are numerical data which express tissue density (attenuation coefficient) in HU and allows its location in space and also refers to a

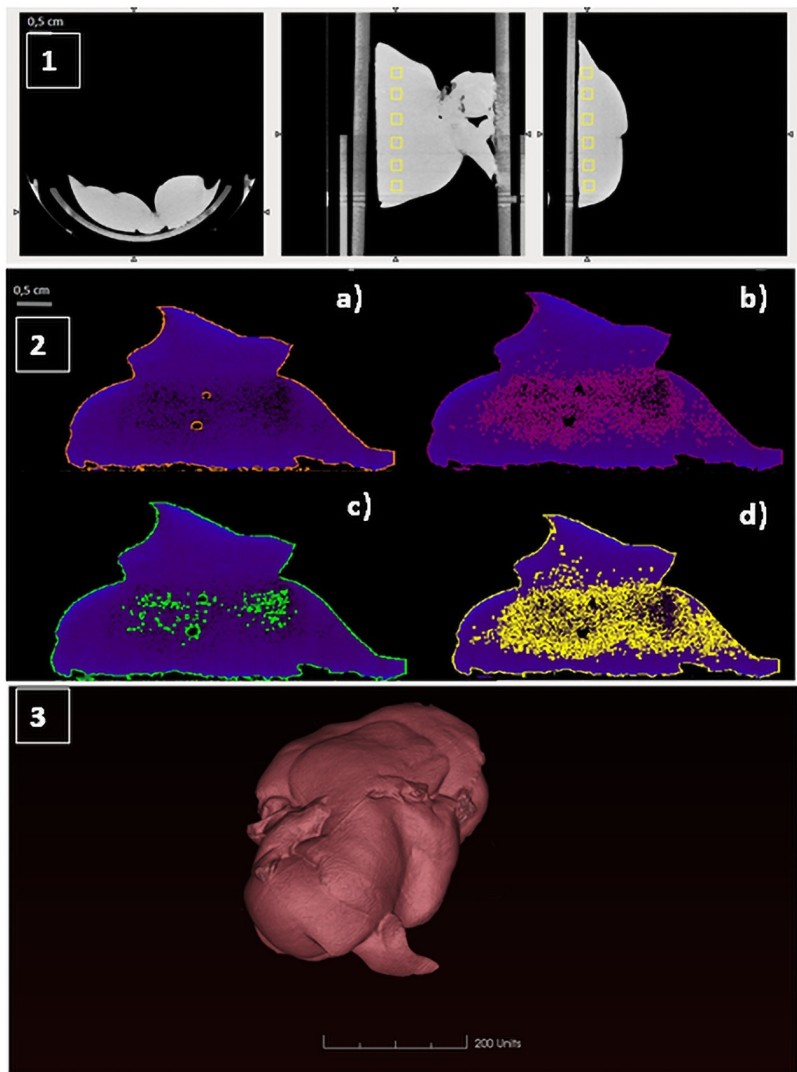

**Fig 3. Micro-TC images of liver samples.** (1) Hepatic parenchyma attenuation coefficient determination measured in HU in six different 8 mm$^3$ volumes of Interest (VOIs) (yellow squares). (2) Hepatic automatic segmentation by AMIDE Software. Segmentation was performed setting attenuation coefficient at 0 (a), 50 (b), 100 (c) and 120 (d) HU.

particular voxel (three-dimensional structure that in this case has a size of 0.125 mm). Our results revealed that, for liver parenchyma, the mean attenuation range was 134–137 HU, while for the liver capsule, values in range 150–200 HU.

Values differing from this range, could reflect an underlying disease or disfunction. For instance, Liver fatty infiltration (hepatic steatosis) could drop hepatic parenchyma density, as fat usually presents lower HU values than healthy parenchyma. In the case of iron overload or cirrhosis, may rise liver density and therefore, HU values. Moreover, the segmentation method used to analyse liver parenchyma density minimizes interoperator biases, since it does not fully depend on the expertise of the radiologist.

The methodology and results presented in this work could have a positive impact in the 3Rs principles application. In this way, it could help to refine liver function evaluation methods, as well as to reduce the number of animals needed. Furthermore, it could help to build a

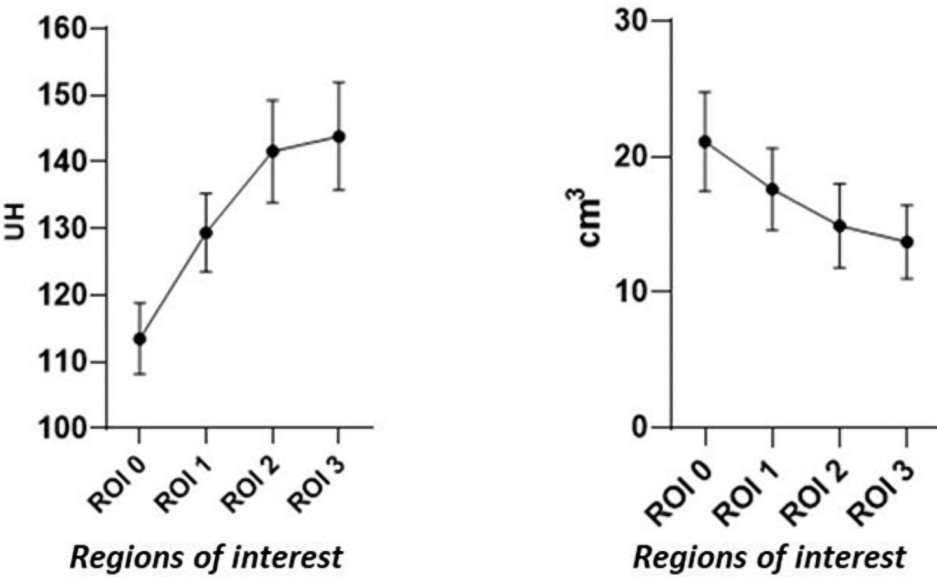

**Fig 4. Hepatic average volume for all rats and the average attenuation coefficient.**

bidirectional bridge between basic and applied researches, adapting clinical equipment and protocols to animal models studies. The methodology and results presented in this work could have a positive impact in the 3Rs principles application. In this way, it could help to refine liver function evaluation methods, as well as to reduce the number of animals needed. Furthermore, it could help to build a bidirectional bridge between basic and applied researches, adapting clinical equipment and protocols to animal models studies.

This is a preliminary study with some limitations. This work only included 6 male SDs rats, with an age ranged between 12 and 16 weeks. In order to minimize variability and stablish differences between physiological parameters, a large number of animals, including females, as well as a wide range of age, should be considered for future studies.

It would be of great value to confirm the ranges of normality to use a pathological model. This objective has already been raised by our research group, but given the variety of parameters and techniques that we wanted to use in this preliminary study, the idea was to determine what type of determinations could be more sensitive to changes according to the pathology studied, by For example, situations of acute or chronic liver disease, in the determination of a fatty liver, assess biochemical parameters, but especially ultrasound and tomography, to see the sensitivity of the technique (HU assessment) to the histopathological changes that occur in this type of tissues.

## Conclusions

This study tries to elucidate reference values for assessing SD normal hepatic function and perfusion. Publications in this regard are scarce, but many of the published data are in concordance with the ones presented in this study. Its establishment would allow minimizing the number of animals needed for non-clinical investigations. However, it has to be considerate that some analysed parameters depend on animal age or hour of sampling. Indocyanine green could be used in SD rats as a suitable marker of liver function. The ultrasound characterization of hemodynamic parameters provides valuable information, which would make it possible to

correlate local dysfunctions with their systemic consequences. Micro-CT, through the study of raw data, help to characterize liver parenchyma, and could be a potential tool for early detection of liver parenchymal alterations and linear follow-up of patients. The characteristics of the studied population, allow us to establish these values in which all were healthy livers, being recommended the inclusion of a second group with known liver disease for future studies.

## Supporting information

**S1 Table. Blood biochemical data availability.** Detailed description of each of the values per week.
(XLSX)

**S2 Table. Cardiovascular hemodynamic, hepatic and renal perfusion data availability.** Detailed description of each results per rat.
(XLSX)

**S3 Table. Micro-CT images data availability.** Detailed description of each VOI values at 0, 50, 100 and 120 minutes according to attenuation coefficient and hepatic volume.
(XLSX)

## Author Contributions

**Conceptualization:** Victor Lopez-Lopez, Nuria Garcia-Carrillo, Diego de Gea, Lidia Oltra, Carlos Alberto González-Bermúdez, Roberto Brusadin, Asunción Lopez-Conesa, Ricardo Robles-Campos.

**Data curation:** Victor Lopez-Lopez, Nuria Garcia-Carrillo, Diego de Gea, Lidia Oltra, Carlos Alberto González-Bermúdez, Guillermo Carbonell, Roberto Brusadin, Ricardo Robles-Campos.

**Formal analysis:** Victor Lopez-Lopez, Nuria Garcia-Carrillo, Diego de Gea, Carlos Alberto González-Bermúdez.

**Funding acquisition:** Nuria Garcia-Carrillo, Diego de Gea, Carlos Alberto González-Bermúdez, Guillermo Carbonell.

**Investigation:** Diego de Gea, Lidia Oltra, Guillermo Carbonell, Asunción Lopez-Conesa.

**Methodology:** Victor Lopez-Lopez, Nuria Garcia-Carrillo, Diego de Gea, Lidia Oltra, Carlos Alberto González-Bermúdez, Guillermo Carbonell, Asunción Lopez-Conesa.

**Software:** Nuria Garcia-Carrillo.

**Supervision:** Ricardo Robles-Campos.

**Validation:** Lidia Oltra, Carlos Alberto González-Bermúdez, Guillermo Carbonell, Roberto Brusadin, Asunción Lopez-Conesa, Ricardo Robles-Campos.

**Visualization:** Roberto Brusadin.

**Writing – original draft:** Victor Lopez-Lopez, Nuria Garcia-Carrillo, Diego de Gea, Asunción Lopez-Conesa, Ricardo Robles-Campos.

**Writing – review & editing:** Victor Lopez-Lopez, Nuria Garcia-Carrillo, Lidia Oltra, Carlos Alberto González-Bermúdez, Guillermo Carbonell, Roberto Brusadin, Asunción Lopez-Conesa, Ricardo Robles-Campos.

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
