## [Decision Letter · Decision Letter 0]

27 Apr 2021

PONE-D-20-37951

Assessment of hepatic function, perfusion and parenchyma attenuation with indocyanine green, ultrasound and computed tomography in a rat model: preliminary standardization of baseline parameters in a healthy liver.

PLOS ONE

Dear Dr. Lopez,

Thank you for submitting your manuscript to PLOS ONE. After careful consideration, we feel that it has merit but does not fully meet PLOS ONE’s publication criteria as it currently stands. Therefore, we invite you to submit a revised version of the manuscript that addresses the points raised during the review process.

Please submit your revised manuscript within 3 months. If you will need more time than this to complete your revisions, please reply to this message or contact the journal office at plosone@plos.org. Please include the following items when submitting your revised manuscript:

We look forward to receiving your revised manuscript.

Kind regards,

Mehmet A Orman

Academic Editor

PLOS ONE

Journal Requirements:

3. To comply with PLOS ONE submissions requirements, please provide methods of sacrifice in the Methods section of your manuscript.

Reviewers' comments:

Reviewer's Responses to Questions

**Comments to the Author**

1. Is the manuscript technically sound, and do the data support the conclusions?

Reviewer #1: Yes

Reviewer #2: Yes

Reviewer #3: Partly

2. Has the statistical analysis been performed appropriately and rigorously? 

Reviewer #1: Yes

Reviewer #2: No

Reviewer #3: No

3. Have the authors made all data underlying the findings in their manuscript fully available?

Reviewer #1: Yes

Reviewer #2: No

Reviewer #3: Yes

4. Is the manuscript presented in an intelligible fashion and written in standard English?

Reviewer #1: Yes

Reviewer #2: No

Reviewer #3: Yes

5. Review Comments to the Author

Reviewer #1: The article is written according to the required standards. It does not come with new data from the literature. It is written punctually in connection with the data collected during the experiment and the analyzes collected. Even if it is on a very small batch of experienced animals it comes with clear data to follow. Thus, it can be an article that will be read by researchers who want to put evaluations into practice on the same subject, a condition that leads this article to have its place in the chapter of the experimental guide. Experimental models can indeed be concretized and performed on small batches of animals if everything is very clearly standardized but in the case of this article only 6 is a fairly small number. I believe that the publication of the article will encourage the authors to go further with the experiments for the concretization and finishing of the necessary experimental models.

Reviewer #2: The manuscript showed an interesting finding but many concerns are raised as follows:

1) The experiments were done only by six rats, which are quite low number of animal in the experiments. Also, some statistical analyses were done for only one rat. How?

2) The number of rats and their age were repeated twice in pages 11 (methods) and 15 (results).

3)Th biochemical assays were not supported by reference citation of the methods (page 11).

4) What is Min1, 5 & 10 in Table 3. It should be 1 min, 5 min & 10 min.

5) An editing is required for figures. In Figure 1, what is the name of the Y axis?. In Figure 4, what is the name of X-axis and what is ROI?. Better illustrations are required for Figures 2 & 3.

Finally, the authors should repeat their results on unhealthy rats with liver disease to confirm their results.

Reviewer #3: The manuscript seeks to establish normal reference ranges for rats for non-invasive measures of liver function. These parameters may lead to the reduction of the number of animals used on experiments.

The methods and experiments are sound. The novelty and impact of the study is very low. There are many limitations as discussed by the authors in the conclusions--only males were used, very small number of animals & values for a measurement were thrown out due to laboratory error. The impact on the 3 Rs is likely minimal as many researchers collect tissue at multiple timepoints in order to assess changes at the molecular and cellular level. Also, the technology used and expertise needed is not readily accessible at most institutions.

Major Comments:

1. The blood chemistry values are very different for several outcomes compared to reference ranges from prior studies. No explanation is given for these differences. This should be discussed in the discussion section of the manuscrupt.

2. The purpose of the technology is to provide a reference range for normal. It would greatly enhance the studies to test the change from reference in a disease model.

6. PLOS authors have the option to publish the peer review history of their article (what does this mean?). If published, this will include your full peer review and any attached files.

Reviewer #1: No

Reviewer #2: No

Reviewer #3: No

---

## [Author Response · Author response to Decision Letter 0]

4 Jun 2021

Academic Editor Mehmet A Orman

Murcia, May 25th, 2021

Revisions of manuscript Ms# PONE-D-20-37951-R1:

 "Assessment of hepatic function, perfusion and parenchyma attenuation with indocyanine green, ultrasound and computed tomography in a rat model: preliminary standardization of baseline parameters in a healthy liver“.

Dear ladies and gentlemen of the editorial board of PLOS ONE,

We would like to thank the editorial board and the 3 reviewers for their interest shown in our submitted manuscript and their time and effort to review our study. Their valuable comments and critique are greatly appreciated. We have modified the paper according to the suggestions made; all changes in the text are outlined with red background color. Furthermore, our point-by-point responses to each comment of the reviewers are attached below. 

We hope that all questions were answered to the satisfaction of the Reviewers and the necessary changes implemented. Should any questions remain or further changes to the manuscript deemed necessary, please do not hesitate to contact us.

Reviewer 1

The article is written according to the required standards. It does not come with new data from the literature. It is written punctually in connection with the data collected during the experiment and the analyzes collected. Even if it is on a very small batch of experienced animals it comes with clear data to follow. Thus, it can be an article that will be read by researchers who want to put evaluations into practice on the same subject, a condition that leads this article to have its place in the chapter of the experimental guide. Experimental models can indeed be concretized and performed on small batches of animals if everything is very clearly standardized but in the case of this article only 6 is a fairly small number. I believe that the publication of the article will encourage the authors to go further with the experiments for the concretization and finishing of the necessary experimental models.

We appreciate the comments delivered by reviewers to improve the scientific quality of our manuscript. We described several non-invasive diagnostic methods to evaluate normal liver parenchyma in an attempt to implement these tools in animal research. 

We address several limitations on our study (small sample size and only males) but we would like to publish our results as a pilot study to encourage ourselves and other research groups to keep working on the same direction, and evaluate other pathological models as the reviewers and editor propose.

Reviewer 2

The manuscript showed an interesting finding but many concerns are raised as follows:

1) The experiments were done only by six rats, which are quite low number of animal in the experiments. Also, some statistical analyses were done for only one rat. How?

The parameters related to blood biochemical analysis, ICG plasma clearance determination, high-frequency ultrasound examination and micro-CT technique were analyzed in all the rats. The relation to the comment made by the reviewer that some statistical analyzes were done for only one rat we think is related to an error in the explanation of the results on the Micro-TC images of liver samples.

In the manuscript we wrote “hepatic average volume for each rat were 21.08 ± 3.32, 17.57 ± 2.76, 14.87 ± 2.83 and 13.67 ± 2.45 cm3 with an average attenuation coefficient of 113.51 ± 18.08, 129.19 ± 7.18, 141.47 ± 1.95 and 151 , 67 ± 1.2 HU”. This phrase can give rise to confusion, so we have modified the phrase by “hepatic average volume for all the rats were 21.08 ± 3.32, 17.57 ± 2.76, 14.87 ± 2.83 and 13.67 ± 2.45 cm3 with an average attenuation coefficient of 113.51 ± 18.08, 129.19 ± 7.18, 141.47 ± 1.95 and 151.67 ± 1.2 HU”.

2) The number of rats and their age were repeated twice in pages 11 (methods) and 15 (results).

Following the reviewer's recommendations, we have corrected it in the manuscript.

3) The biochemical assays were not supported by reference citation of the methods (page 11).

Following the reviewer´recommendations we have supported the biochemical assays with 2 new reference citations:

- Lee, G., & Goosens, K. A. (2015). Sampling blood from the lateral tail vein of the rat. Journal of visualized experiments: JoVE, (99), e52766. https://doi.org/10.3791/52766

- Culley DJ, Baxter MG, Yukhananov R, Crosby G. Long-term impairment of acquisition of a spatial memory task following isoflurane-nitrous oxide anesthesia in rats. Anesthesiology. 2004 Feb;100(2):309-14. doi: 10.1097/00000542-200402000-00020. 

4) What is Min1, 5 & 10 in Table 3. It should be 1 min, 5 min & 10 min.

Table 3. «min 1; min 5; min 10» has been rewritten as «1 min; 5 min; 10 min» following reviewer instructions.

5) An editing is required for figures. In Figure 1, what is the name of the Y axis?. In Figure 4, what is the name of X-axis and what is ROI?. Better illustrations are required for Figures 2 & 3.

Following the reviewer's recommendations name of Y-axis (time) in figure 1 has been added as required and legend of Figure 4 has been rewritten in order to clarify meaning of ROI (X-axis).

Finally, the authors should repeat their results on unhealthy rats with liver disease to confirm their results.

Checking the baseline values of the present study respect to a pathological model would be of great value and would confirm the data obtained as normalized for subsequent studies. This objective has already been raised by our research group. We will compare healthy vs different pathological models as future research, like animals with NAFLD, NASH and liver cirrhosis, performing blood parameters, pathological findings and different imaging techniques (US and CT).

Reviewer 3

Reviewer #3: The manuscript seeks to establish normal reference ranges for rats for non-invasive measures of liver function. These parameters may lead to the reduction of the number of animals used on experiments.

Comments:

The methods and experiments are sound. The novelty and impact of the study is very low. There are many limitations as discussed by the authors in the conclusions--only males were used, very small number of animals & values for a measurement were thrown out due to laboratory error. The impact on the 3 Rs is likely minimal as many researchers collect tissue at multiple timepoints in order to assess changes at the molecular and cellular level. Also, the technology used and expertise needed is not readily accessible at most institutions.

As reviewer points out, the impact on the 3 Rs is likely minimal as many researchers collect tissue at multiple timepoints to assess changes at the molecular and cellular level in pathology studies.

We aimed to establish reference values in healthy Sprague-Dawley rats using different blood test and diagnostic imaging techniques as previously published by other authors. This kind of analysis may be helpful in future research projects to minimize the number of animal samples used and to guide future researchers in similar studies. Our next steps will be to develop and implement these non-invasive techniques to assess different animal models and link our findings in animal research to the clinical practice.

Major Comments:

1. The blood chemistry values are very different for several outcomes compared to reference ranges from prior studies. No explanation is given for these differences. This should be discussed in the discussion section of the manuscript.

The differences between our blood test findings with those described in previous research may be due to multiple factors as animal age, nutrition, environment, or genetics. In our manuscript we describe the scenario we followed in the methodology section. This variability is challenging to manage, as animal research is usually limited to the sample size. In future research we will aim to investigate a bigger cohort of animals to confirm our findings. We assume that our manuscript represents a pilot study that would be helpful to build future projects using different non-invasive diagnostic tools and different animal models.

2. The purpose of the technology is to provide a reference range for normal. It would greatly enhance the studies to test the change from reference in a disease model.

We support the reviewer´s statement and we will perform future studies focused on find differences between healthy and pathological models. We consider our current manuscript as a pilot study and we aim to keep working on the same direction. The use of multiple non-invasive techniques, including imaging and blood tests, may be highly interesting to find potential differences that may help us to detect and characterize the hepatic parenchyma. This approach would help us finding potential correlations between techniques that in combination have the potential to enhance detection and prediction of liver disease.

---

## [Decision Letter · Decision Letter 1]

22 Jul 2021

PONE-D-20-37951R1

Assessment of hepatic function, perfusion and parenchyma attenuation with indocyanine green, ultrasound and computed tomography in a rat model: preliminary standardization of baseline parameters in a healthy liver.

PLOS ONE

Dear Dr. Lopez,

Thank you for submitting your manuscript to PLOS ONE. Unfortunately, the 2nd reviewer was not convinced with your response. His main concerns are about the sample size and the validation of the proposed claims with a disease model. If you decide to address these issues, please submit your revised manuscript within 3 months. If you will need more time than this to complete your revisions, please reply to this message or contact the journal office at plosone@plos.org. Please include the following items when submitting your revised manuscript:

We look forward to receiving your revised manuscript.

Kind regards,

Mehmet A Orman

Academic Editor

PLOS ONE

Reviewers' comments:

Reviewer's Responses to Questions

**Comments to the Author**

1. If the authors have adequately addressed your comments raised in a previous round of review and you feel that this manuscript is now acceptable for publication, you may indicate that here to bypass the “Comments to the Author” section, enter your conflict of interest statement in the “Confidential to Editor” section, and submit your "Accept" recommendation.

Reviewer #2: All comments have been addressed

2. Is the manuscript technically sound, and do the data support the conclusions?

Reviewer #2: Yes

3. Has the statistical analysis been performed appropriately and rigorously? 

Reviewer #2: N/A

4. Have the authors made all data underlying the findings in their manuscript fully available?

Reviewer #2: Yes

5. Is the manuscript presented in an intelligible fashion and written in standard English?

Reviewer #2: No

6. Review Comments to the Author

Reviewer #2: The author's response is not convincing especially the pre-last and last comments for the reviewing of the original version.

7. PLOS authors have the option to publish the peer review history of their article (what does this mean?). If published, this will include your full peer review and any attached files.

Reviewer #2: No

---

## [Author Response · Author response to Decision Letter 1]

13 Nov 2021

Academic Editor Mehmet A Orman

Murcia, Oct 15th, 2021

Revisions of manuscript Ms# PONE-D-20-37951-R2:

 "Assessment of hepatic function, perfusion and parenchyma attenuation with indocyanine green, ultrasound and computed tomography in a healthy rat model: preliminary determination of baseline parameters in a healthy liver“.

Dear ladies and gentlemen of the editorial board of PLOS ONE,

We would like to thank the editorial board and the reviewers for their interest shown in our submitted manuscript and their time and effort to review our study. Their valuable comments and critique are greatly appreciated. First of all we wanted to apologize for the delay in the shipment but due to personal problems we have not been able to answer before. We have modified the paper according to the suggestions made; all changes in the text are outlined with red background color. Furthermore, our point-by-point responses to each comment of the reviewers are attached below. The first modified is the title of the manuscript, “Assessment of hepatic function, perfusion and parenchyma attenuation with indocyanine green, ultrasound and computed tomography in a healthy male rat model: preliminary determination of baseline parameters in a healthy liver”. Standard English has been revised and edited again following the recommendations of the reviewers.

We hope that all questions were answered to the satisfaction of the Reviewers and the necessary changes implemented. Should any questions remain or further changes to the manuscript deemed necessary, please do not hesitate to contact us. 

Academic editor comments

Unfortunately, the 2nd reviewer was not convinced with your response. His main concerns are about the sample size and the validation of the proposed claims with a disease model.

As reviewer points out, a low sample size was used in our study but there is a sizeable litera-ture on the reduced sample size use in biomedical experiments, several researchers consider six animals per group as adequate sample size (Kramer M, Font E. Reducing sam-ple size in experiments with animals: historical controls and related strategies. Biol Rev Camb Philos Soc. 2017 Feb;92(1):431-445. doi: 10.1111/brv.12237. Epub 2015 Nov 13. PMID: 26565143; Lenth, 2001; Dell, Holleran & Ramakrishnan, 2002; Devane, Begley & Clarke, 2004; Lewis, 2006; McCrum-Gardner, 2010; Porras N, 2002). Experimental models can indeed be concretized and performed on small batches of animals if everything is very clearly standardized. In our case, the study is performed to provide a rough idea of the standard deviation and the baseline pa-rameters of biochemical, ultrasound and tomographic values of liver function in healthy males Sprague Dawley rats. 

We think this study might be useful for future studies in pathological models that suffer altera-tions in the determinations valued as direct and indirect indicators of various situations that involve alterations in liver tissue. The procedures we outline can make a substantial contribu-tion towards the goal of reducing the number of animals used in biomedical research, because understanding normal values of healthy animals could reduce the number of animals required and maximize the information obtained per experiment.

In other hand, its true many researchers collect tissue at multiple timepoints to assess changes at the molecular and cellular level in pathology studies, nevertheless, we aimed to establish reference values in healthy Sprague-Dawley rats using different blood test and diagnostic imaging techniques as previously published by other authors. This kind of analysis may be helpful in future research projects to minimize the number of animal samples used and to guide future researchers in similar studies. Our next steps will be to develop and implement these non-invasive techniques to assess different animal models and link our findings in animal research to the clinical practice. 

We are aware the technology used is not accessible at most institutions, but in order to improve the clinical translation is essential similar technology as clinical studies in animal model. On top, the translational value of animal models could be further enhanced when combined with emerging alternative translational approaches.

Checking the baseline values of the present study respect to a pathological model would be of great value and would confirm the data obtained as normalized for subsequent studies. This objective has already been raised by our research group. The next step will be comparing healthy vs different pathological models as future research, like animals with NAFLD, NASH and liver cirrhosis, performing blood parameters, pathological findings and different imaging techniques (US and CT).

We support the reviewer´s statement and we will perform future studies focused on find differences between healthy and pathological models. We consider our current manuscript as a pilot study and we aim to keep working on the same direction. The use of multiple non-invasive techniques, including imaging and blood tests, may be highly interesting to find potential differences that may help us to detect and characterize the hepatic parenchyma. This approach would help us finding potential correlations between techniques that in combination have the potential to enhance detection and prediction of liver disease.

Even if it is on a very small batch of experienced animals it comes with clear data to follow. Thus, it can be an article that will be read by researchers who want to put evaluations into practice on the same subject, a condition that leads this article to have its place in the chapter of the experimental guide.

---

## [Editor Report · Decision Letter 2]

3 Dec 2021

Assessment of hepatic function, perfusion and parenchyma attenuation with indocyanine green, ultrasound and computed tomography in a rat model: preliminary determination of baseline parameters in a healthy liver.

PONE-D-20-37951R2

Dear Dr. Victor Lopez-Lopez,

We’re pleased to inform you that your manuscript has been judged scientifically suitable for publication and will be formally accepted for publication once it meets all outstanding technical requirements.

Kind regards,

Mehmet A Orman

Academic Editor

PLOS ONE
---

## [Editor Report · Acceptance letter]

7 Dec 2021

PONE-D-20-37951R2 

Assessment of hepatic function, perfusion and parenchyma attenuation with indocyanine green, ultrasound and computed tomography in a healthy rat model: preliminary determination of baseline parameters in a healthy liver. 

Dear Dr. Lopez-Lopez:

I'm pleased to inform you that your manuscript has been deemed suitable for publication in PLOS ONE. Congratulations! Your manuscript is now with our production department. 

Kind regards, 

on behalf of

Dr. Mehmet A Orman 

Academic Editor

PLOS ONE